

# Feasibility of progressive sit-to-stand training among older hospitalized patients

Mette Merete Pedersen[1,2], Janne Petersen[1,3], Jonathan F. Bean[4,5],
Lars Damkjaer[6], Helle Gybel Juul-Larsen[1,2], Ove Andersen[1], Nina Beyer[7]
and Thomas Bandholm[1,2,8]

[1] Optimized Senior Patient Program (Optimed), Clinical Research Centre, Copenhagen University Hospital, Hvidovre, Denmark
[2] Physical Medicine & Rehabilitation Research-Copenhagen (PMR-C); Department of Physical and Occupational Therapy, Copenhagen University Hospital, Hvidovre, Denmark
[3] Section of Biostatistics, Department of Public Health, University of Copenhagen, Copenhagen, Denmark
[4] New England GRECC, VA Boston Healthcare System, Boston, MA, United States of America
[5] Department of Physical Medicine and Rehabilitation, Harvard Medical School, Boston, MA, United States of America
[6] Department of Rehabilitation, Copenhagen Municipality Health Administration, Copenhagen, Denmark
[7] Institute of Sports Medicine & Musculoskeletal Rehabilitation Research Unit, Bispebjerg Hospital, Copenhagen, Denmark
[8] Department of Orthopaedic Surgery, Copenhagen University Hospital, Hvidovre, Denmark

Corresponding author
Mette Merete Pedersen,
mette.merete.pedersen@regionh.dk

## ABSTRACT

**Background.** In older patients, hospitalization is associated with a decline in functional performance and loss of muscle strength. Loss of muscle strength and functional performance can be prevented by systematic strength training, but details are lacking regarding the optimal exercise program and dose for older patients. Therefore, our aim was to test the feasibility of a progression model for loaded sit-to-stand training among older hospitalized patients.

**Methods.** This is a prospective cohort study conducted as a feasibility study prior to a full-scale trial. We included twenty-four older patients ($\geq$65 yrs) acutely admitted from their own home to the medical services of the hospital. We developed an 8-level progression model for loaded sit-to-stands, which we named STAND. We used STAND as a model to describe how to perform the sit-to-stand exercise as a strength training exercise aimed at reaching a relative load of 8–12 repetitions maximum (RM) for 8–12 repetitions. Weight could be added by the use of a weight vest when needed. The ability of the patients to reach the intended relative load (8–12 RM), while performing sit-to-stands following the STAND model, was tested once during hospitalization and once following discharge in their own homes. A structured interview including assessment of possible modifiers (cognitive status by the Short Orientation Memory test and mobility by the De Morton Mobility Index) was administered both on admission to the hospital and in the home setting. The STAND model was considered feasible if: (1) 75% of the assessed patients could perform the exercise at a given level of the model reaching 8–12 repetitions at a relative load of 8–12 RM for one set of exercise in the hospital and two sets of exercise at home; (2) no ceiling or floor effect was seen; (3) no indication of adverse events were observed.
The outcomes assessed were: level of STAND attained, the number of sets performed, perceived exertion (the Borg scale), and pain (the Verbal Ranking Scale).

**Results.** Twenty-four patients consented to participate. Twenty-three of the patients were tested in the hospital and 19 patients were also tested in their home. All three criteria for feasibility were met: (1) in the hospital, 83% could perform the exercise at a given level of STAND, reaching 8–12 repetitions at 8–12 RM for one set, and 79% could do so for two sets in the home setting; (2) for all assessed patients, a possibility of progression or regression was possible—no ceiling or floor effect was observed; (3) no indication of adverse events (pain) was observed. Also, those that scored higher on the De Morton Mobility Index performed the exercise at higher levels of STAND, whereas performance was independent of cognitive status.

**Conclusions.** We found a simple progression model for loaded sit-to-stands (STAND) feasible in acutely admitted older medical patients (≥65 yrs), based on our pre-specified criteria for feasibility.

## INTRODUCTION

In older hospitalized medical patients, self-reported decline in functional skills is common before and during hospitalization (*Covinsky et al., 2003*; *Brown, Friedkin & Inouye, 2004*; *Boyd et al., 2008*; *Mudge, O'Rourke & Denaro, 2010*; *Oakland & Farber, 2014*; *Zisberg et al., 2015*) and associated with low in-hospital mobility (*Brown, Friedkin & Inouye, 2004*; *Zisberg et al., 2015*); 30–35% experience a decline in the ability to perform Activities of Daily Living (ADL) from admission to discharge (*Covinsky et al., 2003*; *Boyd et al., 2008*) and barely one third of these patients return to their preadmission level within the first year after discharge (*Boyd et al., 2008*).

In healthy older adults, even a few days of experimental immobilization or periods of bed rest can reduce muscle strength and functional performance (*Kortebein et al., 2007*; *Hvid et al., 2010*; *Hvid et al., 2014*; *Coker et al., 2014*). Also, older adults are more sensitive to bed rest inactivity and have an impaired ability to fully recover compared to younger adults (*Kortebein, 2009*; *Hvid et al., 2010*; *Hvid et al., 2014*). Lower activity levels are common among hospitalized older adults (*Pedersen et al., 2012*; *Villumsen et al., 2014*), and are linked to a decline in functional performance and associated with new institutionalization and death (*Brown, Friedkin & Inouye, 2004*; *Zisberg et al., 2015*). Moreover, hospitalization is associated with a subsequent loss of muscle strength (*Alley et al., 2010*), putting hospitalized older adults at a higher risk of losing independence as a consequence of their hospitalization. Maintaining independence is considered the most important health outcome by many older adults (*Fried et al., 2011*). Therefore, preventing inactivity and loss of muscle strength and functional performance during hospitalization may well be a way of preventing loss of independence.

According to recent systematic reviews, loss of muscle strength and functional performance can be prevented by systematic strength training in both healthy and ill older adults (*De Morton, Keating & Jeffs, 2007*; *Kraemer & Ratamess, 2004*; *Liu & Latham, 2009*; *Koopman & van Loon, 2009*; *Stewart, Saunders & Greig, 2014*). Also, strength training initiated during hospitalization can prevent decline in strength and functional performance associated with hospitalization (*Sullivan et al., 2001*; *Suetta et al., 2007*). In addition, beneficial effects of strength training on functional performance are reported among newly discharged older adults and among frail community-dwelling older adults (*Chandler et al., 1998*; *Courtney et al., 2012*). In general, exercise programmes for older hospitalized or community-dwelling adults consist of a range of exercises (*Chandler et al., 1998*; *Siebens et al., 2000*; *Alexander et al., 2001*; *Bean et al., 2004*; *Brown et al., 2006*; *Nolan & Thomas, 2008*; *Courtney et al., 2012*; *Tibaek et al., 2013*; *Abrahin et al., 2014*). Few studies have examined the effect of a cross-continuum program initiated during hospitalization and continued after discharge (*Siebens et al., 2000*; *Brown et al., 2006*). Moreover, these previous studies have experienced problems with compliance (*Siebens et al., 2000*; *Brown et al., 2006*) necessitating the importance of ongoing supervision from trained staff even within the home setting (*Siebens et al., 2000*; *Brown et al., 2006*; *Wall, Dirks & van Loon, 2013*). Additionally, details are lacking regarding the optimal nature and dose of exercise (*De Morton, Keating & Jeffs, 2007*; *Liu & Latham, 2009*; *Steib, Schoene & Pfeifer, 2010*). It appears, though, that higher intensities are superior to lower intensities in older adults (*Nicola & Catherine, 2011*; *Raymond et al., 2013*; *White et al., 2015*).

The ideal exercise program for a hospitalized patient should be feasible to perform within a busy care setting. It should be relatively simple requiring minimal equipment and also address the impairments (poor limb strength) and functional deficits (poor mobility skills) common to hospitalized patients (*Bodilsen et al., 2013*; *De Buyser et al., 2014*). Therefore, we focused upon repeated sit-to-stand exercises, since it meets all of these criteria. Our aim was to test the feasibility of a model for progressive sit-to-stand training among older hospitalized patients. Specifically, we wanted to investigate if the progression model would enable the patients to reach a strength training intensity of 8–12 repetitions maximum (RM) for 8–12 repetitions during hospitalization and shortly following discharge, with no indications of ceiling or floor effects for loading, no indications of adverse events and with acceptable exercise adherence.

## METHODS

### Study design

The study is a prospective cohort study conducted as a feasibility study (*Bowen et al., 2009*; *Arain et al., 2010*; *Abbott, 2014*) to indicate the feasibility of a progression model for loaded sit-to-stands when used as a simple strength training exercise. The study was performed from December 2012 to July 2013. Participants were included to test their ability to perform the progressive sit-to-stand exercise once in the hospital and once in their own homes within the first two weeks following discharge. Inclusion took place at Copenhagen University Hospital, Hvidovre, Denmark. The feasibility study was performed prior to

a full-scale randomized controlled trial (ClinicalTrials.gov-identifier: NCT01964482). All participants were informed about the study verbally and in writing before providing written informed consent. The local ethics committee approved the study (H-2-2012-115). The reporting of the study follows the Strengthening the Reporting of Observational Studies in Epidemiology (STROBE) guidelines for cohort studies (*Von Elm et al., 2014*), and the description of the intervention follows the Template for Intervention Description and Replication (TIDieR) checklist (*Hoffmann et al., 2014*). When we designed the present study, endorsement of registration of all trials was not as prevalent as today, which is why it was not registered. All criteria related to feasibility, however, were pre-specified.

## Subjects

Older medical patients (≥65 yrs) acutely admitted from their own home to the medical services of the hospital, via the emergency department, were included by random sampling. The exclusion criteria were: (1) inability to rise from a chair with help; (2) inability to cooperate in measurements; (3) inability to give informed consent to participate; (4) diagnosis of Chronic Obstructive Pulmonary Disease (COPD) and participation in a COPD rehabilitation program; (5) terminal illness or being in cancer treatment; (6) inability to speak or understand Danish; (7) isolation-room stay; (8) transferral to the intensive care unit; (9) an expected hospitalization of one day or less.

## Procedures

All assessments were performed by two skilled physiotherapists—one with 15 years of experience (the primary investigator, MMP), and one with two years of experience (HGJ). The same physiotherapist performed all assessments for a given patient. Before initiation of the study, HGJ was trained in all assessments and the progression model and assisted MMP in assessing the first two patients to ensure standardization.

## Descriptive data

Medical records were extracted for demographic data, co-morbidities, length of hospital stay, admission diagnosis, and discharge destination. The patients underwent a structured baseline interview within the initial 48 h of the hospital stay, to collect information about marital status, residence before hospitalization, recent weight loss, basic mobility, functional independence, physical activity level 2 weeks prior to admission, health status, nutritional status, cognitive status, and mobility: the Cumulated Ambulation Score (CAS) was used as an objective measure of basic mobility. It quantifies the patients' independence in three basic activities: getting in and out of bed, sit-to-stand from a chair, and walking (*Foss, Kristensen & Kehlet, 2006*); the New Mobility Score (NMS) was used to assess functional independence in retrospect 2 weeks before admission and in retrospect over the day of admission, respectively (*Parker & Palmer, 1993*); the level of self-reported physical activity was assessed by a questionnaire modified by Schnohr (*Saltin & Grimby, 1968*; *Schnohr, Scharling & Jensen, 2003*) categorizing physical activity of the patient in level 1: low physical activity, level 2: moderate physical activity, and levels 3 + 4: high physical activity; The EQ-VAS of the EQ-5D was used to assess health status (*Rabin & de Charro, 2001*); and

Nutritional Risk Screening (NRS) was used to screen for nutritional risk (*Kondrup, 2003*). In addition, two possible modifiers were assessed both on admission and in the patients' own homes: (1) the De Morton Mobility Index (DEMMI) (score 0–100) to quantify the patient's mobility level before performing the exercise (*De Morton, Davidson & Keating, 2008*). A level of <62 is below normative values for community-dwelling older adults and thus considered to reflect limited mobility (*Macri et al., 2012*); (2) The Short Orientation-Memory-Concentration test (OMC) to assess cognitive status (*Katzman et al., 1983*). A score of 0 reflects the worst cognitive status and a score of 28 reflects the best cognitive status. A score ≤22 was considered to reflect impaired cognition (*Wade & Vergis, 1999*).

## The progression model for loaded sit-to-stands (STAND)

We developed a progression model for loaded sit-to-stands as a strength training exercise and named the model STAND (Fig. 1). STAND was intended to be suitable for older medical patients in the hospital and in their own homes and to ensure training to muscular fatigue in both settings. While developing STAND several meetings were held with physiotherapists from the municipality of Copenhagen to include their ideas on the contents of the different levels of the model. Within 48 h of admission, the patients were contacted at the ward by one of the two physiotherapist to test their ability to perform a sit-to-stand strength training exercise for the lower extremities (acute-phase feasibility). On day one or two after discharge from the hospital, the patients were contacted again by telephone to arrange a re-test of the ability to perform the strength training exercise in their own homes (stable-phase feasibility). The difficulty of the exercise was predefined by STAND ensuring exercise to muscular fatigue in every exercise set (Fig. 1). The easiest level of STAND (level 1) was seated knee-extensions with or without a weight-cuff, which simulates some of the muscle actions required to go from sit-to-stand. Weight cuffs of 0.5 kg, 1 kg, 1.5 kg, 2 kg, 3 kg, 4 kg and 5 kg were used. The most difficult level (level 8) was squat on one leg with added extra weight in the form of a weight vest (Titan Box, 30 kg). The vest had 30 pockets, 15 on the front and 15 on the back, each of which could contain a 1 kg weight—the maximal load of the vest being 30 kg.

The patient was seated on a standard chair with armrests, and a seat height of approximately 45 cm. As a warm-up exercise, the patient was asked to perform five unloaded knee extensions for each limb. The starting point in STAND was level 5 (Fig. 1): sit-to-stand with arms crossed over the chest. From at seated position, the patient was asked to rise to a fully extended position and to sit down in a constant pace. The patient was verbally encouraged to perform as many repetitions as possible maintaining the same pace to ensure training to muscular fatigue (*Tan, 1999*). All exercises were performed at a moderate velocity with both the concentric (raising) and the eccentric (lowering) component being performed over two seconds, separated by a one-second isometric pause after the concentric and eccentric phases, respectively (*Kraemer & Ratamess, 2004*). Both sessions (in-hospital and at home) aimed at three sets of 8–12 repetitions maximum (henceforth: 8–12 RM) corresponding to training at 60–70% of 1 RM (*Tan, 1999*; *Kraemer et al., 2002*; *Kraemer & Ratamess, 2004*). In each set, the aim was to reach fatigue at 8–12

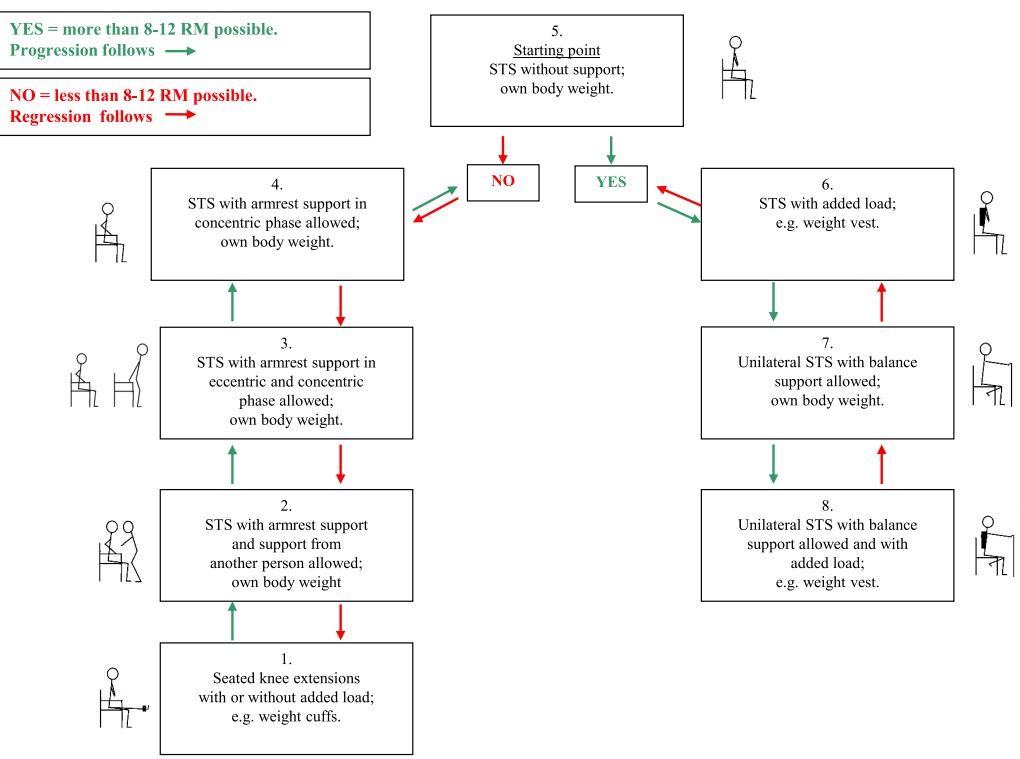

STS: Sit-to-stand; 8-12 RM: 8-12 repetitions maximum (a zone in which muscular fatigue should be reached)

**Figure 1** Progression model for loaded sit-to-stand exercise (STAND). bis. Description of model-procedure.

**Notes.**
**Preparation**
Seated on a standard chair with armrests, and a seat height of approximately 45 cm, the individual should perform 5 unloaded knee extensions for each limb as a warm-up.
**Procedure**
– Perform all exercises at a moderate velocity with both the concentric (raising) and the eccentric (lowering) component being performed over 2 s, separated by a 1-second quasi-isometric pause after the concentric and eccentric phases, respectively.
– Perform as many repetitions as possible maintaining the same pace to ensure training to muscular fatigue.
– If muscular fatigue is reached within 8–12 repetitions, stay at the same level.
– If muscular fatigue is reached before 8 repetitions, perform the exercise at a lower level.
– If muscular fatigue is reached after more than 12 repetitions, perform the exercise at a higher level.
– Aim at 3 sets of 8–12 repetitions to muscular fatigue ($3 \times 8$–12 RM).
– Allow minimal extra support after 6 non-compensatory repetitions to attain muscular fatigue—if a proper technique is maintained.
– Allow increased speed in the last two repetitions if necessary to ensure training at the highest possible level.
– Adjust loads/levels on a set-by-set basis.
– Ensure a 1-minute pause between sets.

*(Continued on next page)*

*(Continued...)*
**Levels—the starting point is level 5:**
All levels are started from a seated position.
Level 1: Attach an appropriate weight cuff ($\geq$0.5 kg) around the ankle. Fully extend the knee and bend it reaching 90° flexion.
Level 2: From a seated position, rise to a fully extended position and sit down using the armrests as support and with additional support from the physiotherapist.
Level 3: From a seated position, rise to a fully extended position and sit down using the armrests as support.
Level 4: From a seated position, rise to a fully extended position using the armrests as support. Sit down with the arms crossed over the chest.
Level 5: From a seated position with arm crossed over the chest, rise to a fully extended position and sit down.
Level 6: From a seated position with arm crossed over the chest and wearing a weight vest (1–30 kg), rise to a fully extended position and sit down.
Level 7: From a seated position (hands on chair in front of you for balance support), rise to a fully extended position on one leg and sit down (shift legs after each set, aiming at 3 sets per leg).
Level 8: From a seated position wearing a weight vest (1–30 kg) (hands on chair in front of you for balance support), rise to a fully extended position on one leg and sit down (shift legs after each set, aiming at 3 sets per leg).

RM (*Kraemer & Ratamess, 2004*), and the correct level of STAND was chosen accordingly (Fig. 1). A two-minute pause was held between sets (*Kraemer & Ratamess, 2004*). In order to ensure that an appropriate training load was achieved, a given level of training was accepted if the patient could perform six non-compensatory repetitions and needed extra support performing the last repetitions (e.g., minimal use of armrests) as long as a proper technique could be maintained. Moreover, increased speed in the concentric phase was allowed in the last two repetitions to optimize limb power output, as leg power has been shown to be associated with physical performance in mobility-limited older adults (*Bassey et al., 1992*; *Bean et al., 2002*). The same skilled physiotherapist supervised all exercise sessions and assessed the level of each patient throughout the sets. The duration of each exercise session was 10–15 min.

## Outcomes measures
### Criteria for feasibility
STAND was considered feasible if three criteria were fulfilled: (1) 75% of the assessed acute-phase patients and stable-phase patients, respectively, could perform the exercise at a given level of the model without session failure. In the hospital, a session failure was defined as inability to perform at least one set of 8–12 RM, and at home a session failure was defined as inability to perform at least two sets of 8–12 RM. One to three sets are recommended for improving muscular strength in older adults (*Kraemer & Ratamess, 2004*) and both one set and multiple sets have been shown to be efficient in improving physical performance and muscle strength in older women (*Abrahin et al., 2014*). Thus, a smaller training volume was accepted in the acute-phase. All causes of session failure were recorded; (2) no clustering of patients at the lowest level (level 1) or the highest level (level 8) was seen—no ceiling or floor effect; (3) no indication of adverse events were observed, e.g., no persistent increase in pain.

### Training level and -load

For each set in the two sessions (in-hospital and at home), the level in STAND, the extra load added (kg), and the number of repetitions were noted.

### The Borg scale

The Borg Scale was administered immediately after each set of the exercise as a measure of perceived exertion (*Borg, 1970*). In healthy older adults, a Borg score of 14–16 has been shown to correspond to 70–90% of 1 RM (*Row, Knutzen & Skogsberg, 2012*) and the Borg score was used as an indicator of whether the perceived effort corresponded with the RM level.

### The Verbal Ranking Scale (VRS)

Before and after assessment of the DEMMI and before, during, and 10 min after the exercise, the patients were asked if they felt pain and wherefrom by the use of the VRS (*Melzack, 1975*). The absence of pain was not a feasibility criterium, but information on pain was collected to gain knowledge about potential adverse events.

## Statistical analysis

No formal sample size calculation was performed due to the descriptive character of the study and as no efficacy testing was to be performed (*Arain et al., 2010*; *Abbott, 2014*). However, a sample size of 24 was decided to be sufficient to obtain a proper variability in the functional level of the patients and thereby be able to evaluate the feasibility of the model in older medical patients. The feasibility results are presented as descriptive data given as means with standard deviations, medians with inter-quartile ranges or percentages, depending on variable type. To evaluate if the level of STAND depended on mobility and cognition, linear regression analyses were used to regress the level of STAND on DEMMI and OMC, respectively. Change in performance measures from admission to at home was tested using Wilcoxon Signed Rank test and the paired *t*-test depending on variable type. All data were double entered in the programme 'Epidata Software' (version 3.1) and all data management and analyses were performed using the SAS version 9.3.

## RESULTS

### Patient characteristics

A total of 248 patients were assessed for eligibility and fulfilled the inclusion criteria. Of these, 200 were excluded based on our exclusion criteria: six were unable to rise from a chair with help; 65 were not able to participate (e.g., due to dementia or confusion); one was participating in a COPD rehabilitation program; 15 were in cancer treatment or terminally ill; four were unable to speak or understand Danish; three were transferred to an isolation room; and 106 were discharged within the first 24 h (Fig. 2). Forty-eight were asked to participate in the study. Of these, 24 patients consented to participate in interviews and tests and 24 declined to participate. The patients were included over a period of 13 weeks with an average inclusion of 1.8 patients per week. One patient dropped out during the initial examination, leaving 23 patients to be tested at the hospital. Two patients did not

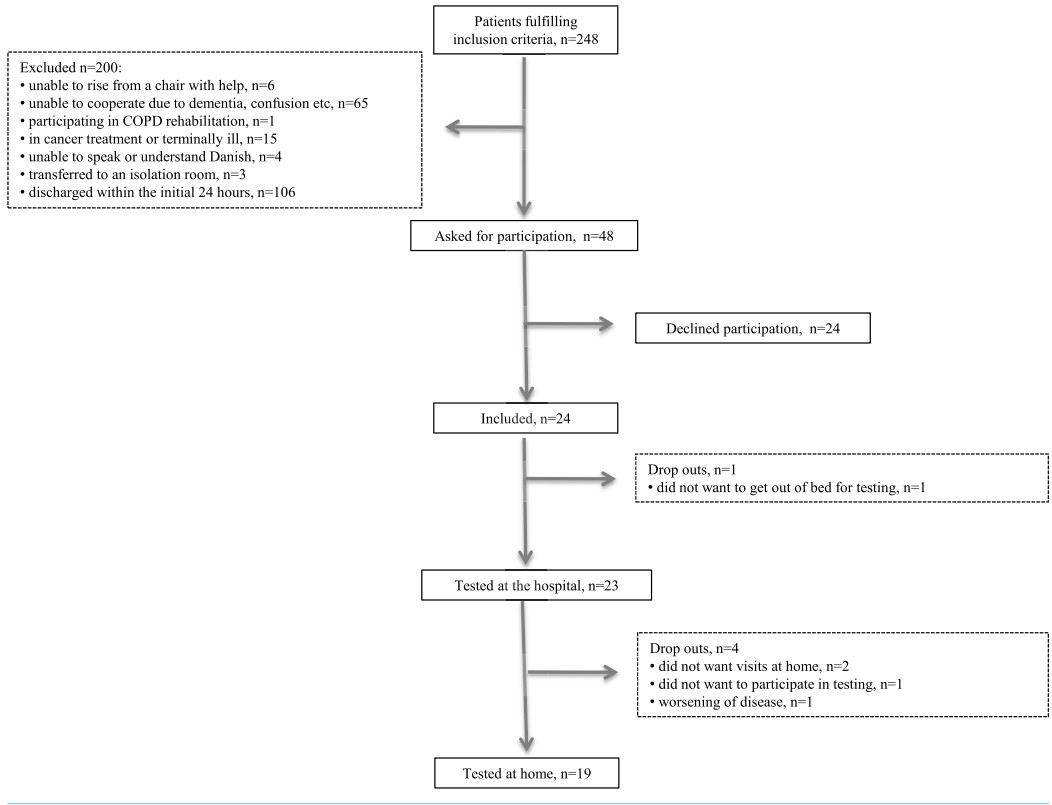

**Figure 2** Flowchart.

want the following home visit, one patient declined to participate in testing at home, and one patient was unable to participate due to worsening of disease, leaving 19 patients to be tested at home. Thus, a total of 20.8% dropped out of the study. Patient characteristics are presented in Table 1. No patients changed in CAS from admission to follow-up. Also, no significant change was seen in NMS and DEMMI whereas self-rated health improved significantly (Table 2).

## Feasibility

### *Sets and loading*

At the hospital, 20 of the 23 patients (83%) were able to perform at least one set of 8–12 RM at a given level of STAND—the remaining three patients stopped after 6–7 repetitions; one due to dyspnea, one due to muscular fatigue, and one due to back pain that was present before performing the exercise. All three patients were subsequently able to perform several sets of 8–12 RM in their own home.

At home, 15 of the 19 patients (79%) were able to perform two sets of 8–12 RM, and 8 of these were able to perform three sets of 8–12 RM. Reasons for not attaining the goal of two sets of 8–12 RM were: one patient could perform seven repetitions in set one and 10 repetitions in set two; one patient stopped after one set due to knee pain—this pain did not persist after ending the exercise; one patient wanted to stop after one set due to a

**Table 1  Patient characteristics on admission.**

|  | N |  |
|---|---|---|
| Age; mean (SD) | 24 | 77 ± 7 |
| Gender, female; $n$ (%) | 24 | 12 (50%) |
| Living alone, yes; $n$ (%) | 24 | 13 (54%) |
| Use of gait devices, yes; $n$ (%) | 24 | 9 (37.5%) |
| Reason for admission; $n$ (%) | 24 | |
| Pneumonia | | 10 (41.7%) |
| COPD exacerbation | | 2 (8.3%) |
| Dyspnea | | 1 (4.2%) |
| Urinary tract infection | | 3 (12.5%) |
| Gastroenteritis | | 1 (4.2%) |
| Pulmonary embolism | | 2 (8.3%) |
| Atrial fibrillation | | 3 (12.5%) |
| Anemia | | 2 (8.3%) |
| Physical activity level (PA); $n$ (%) | 23 | |
| Low PA | | 5 (21.7%) |
| Moderate PA | | 5 (21.7%) |
| High PA | | 13 (56.6%) |
| Comorbidities; $n$ (IQR) | 24 | 5 (3.5;5.5) |
| Medications; $n$ (IQR) | 24 | 6 (2.5;7.5) |
| Length of stay; median (IQR) | 24 | 4.5 (3;7) |
| Follow-up—number of days after discharge; median (IQR) | 19 | 9 (6;13) |
| Nutritional risk screening | 24 | |
| At risk; $n$ (%) | | 19 (79.2%) |
| OMC; median (IQR)/$n$ (%) | 24 | 26 (22;28) |
| CAS; median (IQR) | 24 | 6 (6;6) |
| NMS, 14 days prior to admission; median (IQR) | 24 | 9 (5.5;9) |
| NMS at admission; median (IQR) | 24 | 3 (2;9) |
| DEMMI; mean (SD) | 23 | 66.1 ± 15.18 |

**Notes.**

OMC, The Short Orientation-Memory-Concentration test; CAS, The Cumulated Ambulation Score; NMS, The New Mobility Score; DEMMI, The De Morton Mobility Index.

**Table 2  Performance measures on admission and at home.**

| Performance measure | N | Admission | N | Home-visit | $P$-value |
|---|---|---|---|---|---|
| CAS; median (IQR) | 24 | 6 (6;6) | 20 | 6 (6;6) | NA[a] |
| NMS admission; median (IQR) | 24 | 3 (2;9) | 20 | 6.5 (3;9) | 0.13 |
| DEMMI; mean (SD) | 23 | 66.1 (15.18) | 19 | 70.6 (14.7) | 0.12 |
| EQ-VAS; mean (SD) | 24 | 56.6 (24.3) | 20 | 67.4 (23.8) | 0.01 |

**Notes.**

[a] No participants changed in CAS.

sensation of muscular fatigue during the first set; one patient wanted to stop in set two due to a sensation of muscular fatigue.

The 20 patients completing one set at the hospital were distributed in STAND as follows: two seated knee extensions, two sit-to-stand using the arm rests when standing and sitting down, two sit-to-stand using the arm rests when sitting down, six sit-to-stand with the arms crossed over the chest, six sit-to-stand with extra load, one unilateral sit-to-stand, and one unilateral sit-to-stand with extra load. The 15 patients completing two sets at home were distributed in STAND as follows: three sit-to-stand using the arm rests when standing up and sitting down, one sit-to-stand using the arm rests when sitting down, four sit-to-stand with the arms crossed over the chest, four sit-to-stand with extra load, one unilateral sit-to-stand, and two unilateral sit-to-stand with extra load (Table 3). The mean Borg score when performing the highest level possible was 14.2 (±1.9) on admission and 14.1 (±1.6) at follow-up.

### Indicators of floor/ceiling effect

Two patients were at the lowest level of STAND at the hospital (knee-extensions with three and six kg, respectively). For both patients, further regression was possible by using less weight (they both performed the exercise at level 3 at home). One patient was at the highest level of STAND at the hospital and two were at the highest level at home (unilateral sit-to-stand with six kg and four kg, respectively)—for both patients, further progression was possible by adding more weight.

### Pain

Four patients and two patients, respectively, reported an increase in pain after the DEMMI test at the hospital and at home. None of these patients reported any pain before the exercise.

Four patients reported light to moderate pain in the shoulder, leg and chest, respectively, before performing the exercise at the hospital. The pain remained unchanged during and after the exercise for three of the patients and one patient reported no pain after ended exercise. Three patients reported light leg pain during the exercise but no pain before and after the exercise. Four patients reported light to moderate pain in the shoulder, back, leg and head, respectively, before performing the exercise at home. The pain remained unchanged during and after the exercise for three of the patients and one patient reported less pain after ended exercise. Two patients reported light back pain during the exercise but no pain before and after the exercise.

### Mobility and cognition

As shown in Fig. 3 those that scored higher on the DEMMI performed the exercise at the most challenging levels of STAND (on admission, $\beta = 0.10$ (CI [0.07–0.13]), $P < 0.0001$; at home, $\beta = 0.07$ (CI [0.03–0.12]), $P = 0.004$), whereas the level of STAND did not depend significantly on OMC (on admission: $0.07(-0.12;0.26)$, $P = 0.45$; at home: $-0.01(-0.42;0.41)$, $P = 0.96$).

**Table 3** Overview over the 8 levels of the STAND model and the distribution of patients on the 8 levels according to the highest level performed in the hospital and at home, respectively.

| Level in STAND | Description of level | Illustration | In hospital (n) | At home (n) |
|---|---|---|---|---|
| 1 | Seated knee extensions with or without added load, e.g., weight cuffs. | | 2 | 0 |
| 2 | STS with armrest support and support from another person allowed; own body weight. | | 0 | 0 |
| 3 | STS with armrest support in eccentric and concentric phase allowed; own body weight. | | 2 | 3 |
| 4 | STS with armrest support in concentric phase allowed; own body weight. | | 2 | 1 |
| 5 Starting point | STS without support; own body weight. | | 6 | 4 |
| 6 | STS with added load; e.g., weight vest. | | 6 | 4 |
| 7 | Unilateral STS with balance support allowed; own body weight. | | 1 | 1 |
| 8 | Unilateral STS with balance support allowed and with added load; e.g., weight vest. | | 1 | 2 |

**Notes.**
STS, sit-to-stand.

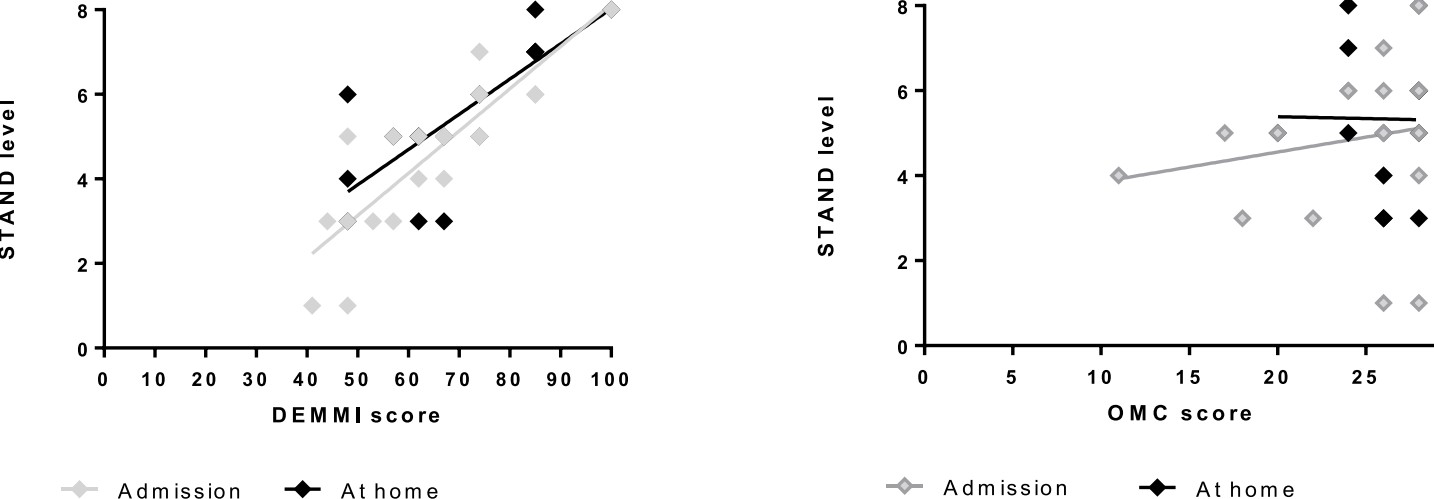

**Figure 3 The association between DEMMI score (A) and OMC score (B), respectively, and performed level of STAND on admission and at home.** DEMMI score: score on the De Morton Mobility Index (0–100). The higher the score the better mobility. OMC score: score on the Short Orientation-Memory-Concentration test (0–28). The higher the score the better cognition. STAND level: 1 indicates lowest level of the model (seated knee-extensions) and 8 indicates highest level of the model (unilateral sit-to-stand with added load).

## DISCUSSION

The major finding of our feasibility study was that our exercise model of progressive sit-to-stands (STAND) was feasible among hospitalized older adults and demonstrated potential for being used in a future study appropriately powered to evaluate the effect of the exercise on mobility, physical activity, functional performance and independence in this population. Specifically, we found that more than 75% of the patients assessed during hospitalization and shortly following discharge in their own home were able to perform the sit-to-stand exercise at a given level of STAND reaching an intensity of 8–12 RM for 8–12 repetitions. No clustering of patients at the highest or lowest level of STAND was seen, suggesting no ceiling or floor effect, and for all patients assessed a possibility of either progression or regression was possible. Finally, no adverse events were reported.

Consistent with this study, previous studies have found resistance training to be feasible in older hospitalized patients (*Siebens et al., 2000*; *Mallery et al., 2003*). However, these studies have used either low intensity exercises; due to a concern of potential risks of exercising older hospitalized patients (*Siebens et al., 2000*); or exercises performed lying in bed (*Mallery et al., 2003*). Our study shows that a performance-based, higher-intensity exercise is feasible both in hospitalized older adults with high and low mobility (*Macri et al., 2012*) (a DEMMI score of 44–80) and with and without mild cognitive impairment (*Katzman et al., 1983*) (an OMC score of 18–28). Moreover, we found a strong association between the level of STAND and DEMMI which indicates that the achieved level of STAND reflected the mobility level of the patients. Additionally, the level of STAND was not associated with cognition, which implies that STAND can be used independent of cognitive level. It has previously been shown that high intensity resistance training is superior to low intensity in frail older adults (*Seynnes et al., 2004*), which is why STAND may be a good

choice in older hospitalized adults. We were able to provide optimal resistances with the exercise as more than 75% of the assessed patients were able to perform the exercise with a loading of 8–12 RM for 8–12 repetitions for the intended number of sets. Of those not able to reach the intended loading/number of sets two thirds stopped after 6–7 repetitions or due to muscular fatigue. This may indicate that they were able to perform the exercise but needed better adjustment of the load or needed better information regarding the management of muscular fatigue when performing strength training. The mean Borg score when performing the highest level possible was 14, corresponding to a 75% effort (*Avers & Brown, 2009*). Thus, this subjectively perceived effort corresponds well with 8–12 RM (*Kraemer & Ratamess, 2004*) and indicates that the patients have exercised at the intended level. Also, no adverse events were seen. Therefore, this mode of progressive exercise seems appropriate as a simple strength training exercise in acutely admitted older medical patients.

## Limitations and strengths

A limitation of the study is that the assessed patients represent a select group of acutely admitted older medical patients as 90% of the patients fulfilling the inclusion criteria were either excluded (80%) or declined to participate (10%). The proportion of patients consenting to participate, however, is equal to (*Mallery et al., 2003*) or higher (*Siebens et al., 2000*; *Brown et al., 2006*) than seen in previous exercise studies in older hospitalized adults, which underlines the difficulty of including patients in the acute setting and limits the generalizability to acutely admitted older patients equivalent to our sample. In addition, we consider our exclusion criteria reasonable as the majority of those excluded either would probably not have been able to perform the exercise with the intended quality (e.g., due to dementia or confusion; 32.5%), or would not benefit from a program including the exercise (e.g., due to being in cancer treatment or terminally ill; 7.5%) or had a very short hospital stay (discharged within the first 24 h; 53%). However, patients excluded due to inability to rise from a chair might benefit from exercise based on the STAND model (level 1) or other interventions based on less demanding exercises equivalent with the ones used by Mallery and co-workers (*2003*). Another limitation of our study is that the feasibility of STAND has only been tested for one session in each setting (hospital and home) and therefore, we are not able to evaluate whether the patients can comply with the exercise over time or whether STAND is sufficient in ensuring the right load over time, e.g., a training period of 4 weeks. We do believe, though, that the model can be used for a longer training period, as progression and regression was possible for all levels of the model and neither floor nor ceiling effect was seen.

A major strength of our study is that the exercise, following STAND, is well-described, simple and low in cost making it possible to implement both in an acute hospital ward as well as in the patients' homes. A study by *Sullivan et al. (2001)* in hospitalized frail elderly showed that 10 weeks of resistance training consisting of three sets of eight leg presses in a leg press chair increased strength and lowered sit-to-stand time. The sit-to-stand exercise (level 2–8 of STAND) corresponds well with the leg press exercise, requiring the

use of similar muscle-synergies. However, in the hospital and especially in the home setting weight-lifting equipment like a leg press chair is not often available why it is promising that using a weight vest and the sit-to-stand exercise patients can be loaded to the same extend enabling low technology resistance training both in the hospital and at home. Additionally, as expressed in several recent reviews it is very important to use exercise programs that are detailed with regard to technique, dosage and progression of the exercise. Our program complies with the recommendation (*De Morton, Keating & Jeffs, 2007*; *Liu & Latham, 2009*; *Steib, Schoene & Pfeifer, 2010*; *Kosse et al., 2013*; *Giné-Garriga et al., 2014*; *Timmer, Unsworth & Taylor, 2014*; *White et al., 2015*). Moreover, the inclusion of physiotherapist supervision ensures optimal dosage and technique and may also enhance compliance. This design element was included to overcome challenges within previous studies that used unsupervised training in the home setting (*Siebens et al., 2000*; *Buhl et al., 2015*).

### Perspective

We are now conducting a randomized controlled trial to test a cross-continuum strength training intervention in older medical patients (NCT01964482). The goal of the trial is to investigate the effect of a simple, supervised strength training program consisting of two lower-extremity strength training exercises. The exercises are based on STAND and performed during hospitalization and the first four weeks after discharge at home.

## CONCLUSIONS

Based on our pre-defined criteria for feasibility we found that a simple progression model for loaded sit-to-stands (STAND) was feasible in acutely admitted older medical patients (+65 yrs) in the hospital- and home setting. Following the progression model, a strength-training intensity of 8–12 RM for 8–12 repetitions was reached for two thirds of the assessed patients with no indication of ceiling or floor effect for load, and no report of adverse events.

## ACKNOWLEDGEMENTS

Thanks to the physiotherapists in the municipality of Copenhagen for helping in the development of the model.

### Funding

This study was funded by the Lundbeck Foundation, Hvidovre Hospital, Danish Regions/The Danish Health Confederation, and The Association of Danish Physiotherapists. The funders had no role in study design, data collection and analysis, decision to publish, or preparation of the manuscript.

### Grant Disclosures

The following grant information was disclosed by the authors:
Lundbeck Foundation.
Hvidovre Hospital.

Danish Regions/The Danish Health Confederation.
The Association of Danish Physiotherapists.

## Competing Interests

The authors declare there are no competing interests.

## Author Contributions

- Mette Merete Pedersen conceived and designed the experiments, performed the experiments, analyzed the data, wrote the paper, prepared figures and/or tables, reviewed drafts of the paper, interpreted analyzed data.
- Janne Petersen conceived and designed the experiments, analyzed the data, contributed reagents/materials/analysis tools, reviewed drafts of the paper, interpreted analyzed data.
- Jonathan F. Bean reviewed drafts of the paper, interpreted analyzed data.
- Lars Damkjaer conceived and designed the experiments, reviewed drafts of the paper.
- Helle Gybel Juul-Larsen performed the experiments, reviewed drafts of the paper.
- Ove Andersen contributed reagents/materials/analysis tools, reviewed drafts of the paper, interpreted analyzed data.
- Nina Beyer and Thomas Bandholm conceived and designed the experiments, reviewed drafts of the paper, interpreted analyzed data.

## Human Ethics

The following information was supplied relating to ethical approvals (i.e., approving body and any reference numbers):

The Ethics Committee of the Capital Region of Copenhagen (H-2-2012-115).

## Data Availability

Raw data can be found in the Supplemental Information.

## Supplemental Information

Supplemental information for this article can be found online at http://dx.doi.org/10.7717/peerj.1500#supplemental-information.

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
