# Peer review of "Feasibility of progressive sit-to-stand training among older hospitalized patients"

_PeerJ, doi:10.7717/peerj.1500_

## Round 0.1 · original submission · Minor Revisions

The manuscript provides valuable and practical information on implementing research in an acute care setting. Other than the points made by the reviewers, I feel that the fact that only 48 patients met inclusion criteria while over 200 patients were excluded reflects the feasibility of the intervention. This suggests that other interventions may be required for those with more complex needs or higher disability. This fact should be addressed in the discussion.

·

Basic reporting

Basic reporting is sound with the exception of the abstract. This is unclear in most parts unless you have read the whoöe manuscript. The abstract has to be revised in total.

Experimental design

Design and other methods are clearly described with only minor revision needed.Since this is a

Validity of the findings

Since this is a feasibility/pilot study, some limitations are given, but are discussed appropriately. Conclusions are drawn from the results and perspectives are given.

Additional comments

Feasibility of progressive sit-to-stand training among older hospitalized patients
#2015:10:7121:0:0:REVIEW

This manuscript describes a new method how to exercise sit-to-stand performance in older hospitalized and functionally disabled persons.

Abstract
In general, most parts are clear not before reading the whole manuscript. Therefore, please revise in total.
What is a “medical” patient? Suppose to delete the term “medical” here.
Methods are not clear. “… to contraction failure (8-12 RM) …” Did you perform an 8-12 RM test? What was the test? Performing STS at one of the 8 levels?
Were cognition and de Morten secondary outcomes?
“The model was considered feasible” I would expect here anything regarding your stated main outcomes
“… one set in the hospital and two sets at home;” what do you mean here?
What about the typical outcomes of feasibility, such as participation, adverse effects, etc.?
Unclear results: 19 patients were not at all tested in the hospital?
79% were not able to perform the lowest level at an 8-12 RM test in the hospital?
Adverse events should belong to the first reported results.

Introduction
“medical patients”?
LL 62-66: I understand that the decline of performance is due to bed rest or immobilization, right? You may add this here.
LL 107-109: Again, this is unclear. You want to know, if any of the 8 levels of STAND can be performed 8-12 times?

Methods
Is there a rationale for excluding COPD patients?
I use the SOMC score quite often and I know that high scores indicate cognitive impairment. Please check your statement (<22 = cognitive impaired).
L 175: again, what do you mean with ”contraction failure”?
L 183: AHHH!! You should give this explanation at the first occasion and I would avoid it in the abstract.
LL 190-211: here you explain very clear! Only, I understand that you sometimes mean repetition, but not RM. This is crucial in Fig 1. If 8-12 repetitions are possible (not RM possible), the progression follows, right? Please check throughout the manuscript.
LL 215-225: What about a measure of adherence in hospital and after discharge?
Is there a rationale to apply parametric and non-parametric statistics? If not, I suppose to use either parametric or non-parametric statistics only, based on sample size (rather non-parametric) and distribution of parameter values.

Results
As I understand, only 48 patients fulfilled your inclusion criteria, because 224 patients had to be excluded based on pre-defined exclusion criteria.

Discussion
LL 329-330: You could add here physical activity, functional performance and independence as possible study endpoints, which you have raised in your introduction.
LL 381-383: What you state here is not correct for those levels in need of ankle weights or the weight vest. This rather could be a limitation.
Also the supervision needed may be a limitation, at least for the home exercises.

·

Basic reporting

See attached doc

Experimental design

See attached doc

Validity of the findings

See attached doc

---

## Round 0.2 · accepted · Accept

The authors have done an excellent job of addressing the reviewer's concerns